# Land Use Function Transition and Associated Ecosystem Service Value Effects Based on Production–Living–Ecological Space: A Case Study in the Three Gorges Reservoir Area

**Fangjie Pan** [1], **Nannan Shu** [2], **Qing Wan** [1,*] and **Qi Huang** [3]

1 School of Management, Wuhan Institute of Technology, Wuhan 430205, China
2 Wuhan Agricultural Technology Extension Center, Wuhan 430012, China
3 Institute of Economic Research on Changjiang River Valley, Hubei Academy of Social Sciences, Wuhan 430077, China
* Correspondence: wanqing@wit.edu.cn

**Abstract:** The transition of land use function and its effects on ecosystem services is a key issue in eco-environmental protection and is the basis of territorial space governance and optimization. Previous studies have typically selected land use types to evaluate ecosystem service value (ESV) and have overlooked comprehensive characteristics of ecosystem services and the mutual feedback relationship between human social systems and the ecosystem. Taking the Three Gorges Reservoir Area, Hubei section (TGRA-HS) as a case study, we used a transition matrix, the revised ESV method, and an ecological contribution rate model to explore land use function transition (LUFT) and its effects on the change in ESV based on the production–living–ecological space (PLES) classification system. The results show that: (1) The transition of land use function based on PLES was the mapping of the evolution of the human–nature relationship in the spatial pattern, which reflected the evolution of the spatial pattern caused by human interference with the continuous development of society; (2) The evolution of PLES showed the characteristics of a reduction in production space (P-space), and an expansion in living space (L-space) and ecological space (E-space). The distribution pattern of PLES from 1990 to 2020 was basically the same, and the characteristics of structural transform reflected the characteristics of project construction in different phases; (3) The E-space contributed the most to the total ESV, and it has risen by CNY $13.06 \times 10^8$. The transition of land use function caused by human construction projects impacts the spatiotemporal change in the regional ESV; (4) The change in ESV induced by LUFT revealed the whole dynamic process of the positive and negative effects of human construction projects on ecosystem services, and the two effects offset each other to keep the ESV relatively stable. The transition of E-space to P-space had the greatest impact on the reduction in ESV, whose contribution rate was 82.76%. The dynamic changes in land use function and ESV corresponding to the different stages of the Three Gorges Project's (TGP) construction reveals the important driving effect of human activities on ecosystem services. It reminds us that humans should not forget to protect the eco-environment when obtaining services from the ecosystem.

**Keywords:** production–living–ecological space (PLES); land use function transition (LUFT); ecosystem service value (ESV); Three Gorges Reservoir Area (TGRA); Three Gorges Project (TGP)

## 1. Introduction

Land resources are the essential element and spatial carrier for human survival and development, and all human activities depend on land resources [1–3]. While economic and social developments have allowed for great achievements and promoted the profound transformation and spatial reconstruction of land use [4,5], this has brought about a lot of noticeable environmental problems, such as the overextraction of natural resources, water and soil loss, a decrease in biodiversity, and the degradation or loss of ecosystem functions [6–10], resulting in significant changes to ecosystem service functions. The most

severe problem is an unbalanced PLES in terms of its spatial structure and function, leading to severe challenges and crises in the sustainable development of territorial space. The important performance of land use transition is the dynamic process of the quantitative and spatial reallocation of limited land resources among different leading functions [11], thereby realizing the transition of the recessive form of land use function [12,13]. The transition of land use function is one of the important changes in land use in recessive patterns. That is, the dynamic process of quantitative re-proportion and spatial reallocation of land resources among production, living, and ecological functions reflect the different stages of regional economic and social transformation [14]. Land use transition not only causes a change in natural landscapes, but also causes a change in biodiversity, ecosystem service functions, and their stability, which threatens human health and ecological security [15–17]. The research on land use transition and its ecological effects has attracted increasing attention [18–21], and it has become an important research topic in current geography, landscape ecology, and ecological economics [22–25].

The LUFT is the further deepening and application of the land use theory in the research of territorial space in the new era. The existing research shows that land use has comprehensive production, living, and ecological functions, and it can be divided into PLES on the basis of the leading functions and use types [26,27]. The theory of PLES was proposed based on the view of element–structure–function in systematic [28]. The PLES was a comprehensive division method of territorial space, and the coordinated development of PLES will produce a synergistic effect where the total function is greater than the sum of its parts. Therefore, the PLES can reflect the complex characteristics of ecosystem services well, and its layout and evolution have a profound impact on the regional ecosystem services. Ecosystem service function is closely related to human welfare, and it is an important basis for human survival and development [29–31]. Under the influence of intrinsic and extrinsic factors [32], the changes in the spatial element and structure of PLES will lead to the transition of their functions, and the transition of the leading functions reflects the different stages of the evolution of regional ecosystem services. Essentially, the eco-environmental problem caused by the LUFT is due to the imbalance of PLES function. Therefore, based on the leading function classification system of PLES, the regional LUFT and the evolution of ecosystem service functions can be linked, giving a new perspective on studying the mutual feedback relationship between human social systems and ecosystems [33–35].

At the present stage, the research on LUFT and its regional ecological effects from the perspective of PLES is still in its infancy. Some researchers have analyzed the LUFT and its eco-environmental effects but have mainly focused on regions, basins, provinces, cities, and counties [36–40]. Furthermore, some researchers have also discussed it from different perspectives [20,41]. However, the existing research is mostly limited to the administrative areas undergoing rapid urbanization. Research on the LUFT in an important ecological function area has received less attention, and its effects on changes in ESV based on the PLES classification are still insufficiently studied [7]. Meanwhile, the ecological characteristics of the PLES classification system are not considered to be granular enough, and the internal connection between the production–living–ecological functions and ecosystem service function is ignored.

The TGRA is an important ecological function area that plays a vital role in ensuring the downstream ecological security and maintaining the ecosystem health of the Yangtze River [42]. The TGRA is also one of the most sensitive and vulnerable areas of the eco-environment in China, and the PLES competition is becoming increasingly fierce here. As the reservoir head of the TGRA, the Hubei section is most rapidly and directly affected by the project, and the issue of LUFT deserves much more attention. Therefore, it is of important theoretical and practical significance to explore the spatiotemporal characteristics of LUFT and its effects on the change in ESV based on the conception and identification of PLES in the TGRA-HS according to the construction stage division of the TGP.

Based on the above-mentioned state of research, this study aims to quantitatively analyze the structure and spatial characteristics of LUFT in the national key ecological

function area and its effects on changes in ESV from the perspective of PLES during the construction of the TGP from the project demonstration to the full operation stage. The contributions of this study are that: (1) The transition of land use function and the PLES are combined to analyze LUFT and its effects on ecosystem services, and it might provide a deeper understanding of the mutual feedback relationship between human social systems and ecosystems, especially the harmony between human development and biodiversity protection; (2) Taking the TGRA-HS as the study area, we analyzed the spatiotemporal evolution and structural transformation of PLES, and the assessment model was established and revised to analyze the quantity and spatial characteristics of ESV change; (3) This study also provides a scientific reference and practical guidance to support the optimization of the spatial development and eco-environmental protection patterns in the TGRA-HS.

## 2. Study Area

The TGRA-HS is located in the middle reaches of the Yangtze River in the southwest of Hubei Province (30°14′–31°34′ N, 110°06′–111°40′ E) and includes three counties (Badong, Zigui, and Xingshan counties) and five districts (Yiling, Dianjun, Xiling, Wujiagang, and Xiaoting districts) (Figure 1). It has a population of about 2.82 million and an area of about $1.21 \times 10^4$ km$^2$. The Hubei section is the reservoir head of the TGRA, and the landform types are complex and diverse, with more mountains and fewer plains. The mountains are high, the slopes are steep, and the valleys are deep. The climate is affected by topographic fluctuations, and the natural vegetation has the characteristics of vertical zoning. The dam site of the TGP is located in Sandouping Town, south-west of Yichang City, Hubei Province, China. The Gezhou dam, which is the regulating dam of the Three Gorges Dam, is located 38 km downstream. Since 1990, with the continuous advancement of the TGP and various ongoing construction projects, the PLES in the TGRA-HS have changed significantly and human activities have strongly interfered with the regional eco-environment. Under the dual drive of policy and economy, the land use pattern of the TGRA-HS has undergone significant changes, and the land use function has also undergone a transformation. In addition, due to the phased implementation of the TGP, the main projects have caused distress, such as the resettlement of immigrants, the submersion of water storage, and the relocation of facilities. This has further intensified the complexity of the study of regional ecological effect changes. The spatiotemporal characteristics of LUFT and the characteristics of the eco-environmental response show significant differences before and after the dam's construction [43,44].

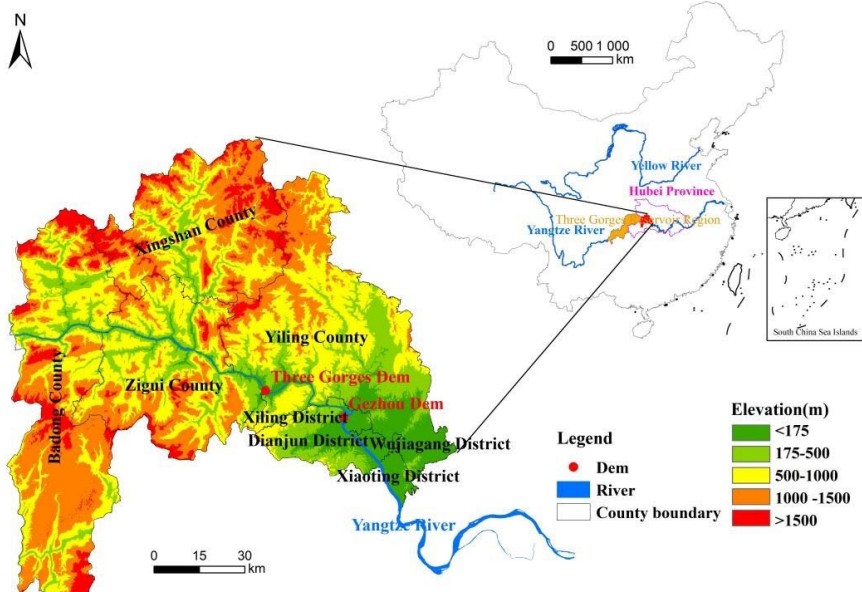

**Figure 1.** Location and elevation of the Three Gorges Reservoir area, Hubei section (TGRA-HS), China.

## 3. Materials and Methods

### 3.1. Research Framework

Different types of land use have various functions. Analyzing the evolution of spatial form and the structure of territorial space is an important way of exploring the process of space function change. One of the manifestations of the land use transition is the transformation between the three leading functions of production, living, and ecology [45]. Under different leading functions (spatial types), human beings have different impacts on the ecosystem, and the services provided by the ecosystem for human society are also significantly different. In general, anthropogenic interference (such as a large-scale construction project) will lead to the transformation of the three leading functions. The structure of PLES tends to be unbalanced, and thus, affects the components, structure, service value, and function of the ecosystem. It leads to changes in the supply capacity of regional ecosystem services and becomes the most direct driving factor affecting the function and value of ecosystem services [46,47]. This study was carried out according to the logic of the construction stage division of the TGP, the characteristics of LUFT, the spatiotemporal change in ESV, and the effects of LUFT in terms of changes in ESV (Figure 2). Thus, a framework of LUFT and its effects in terms of changes in ESV from the perspective of PLES in the TGRA-HS was developed as follows:

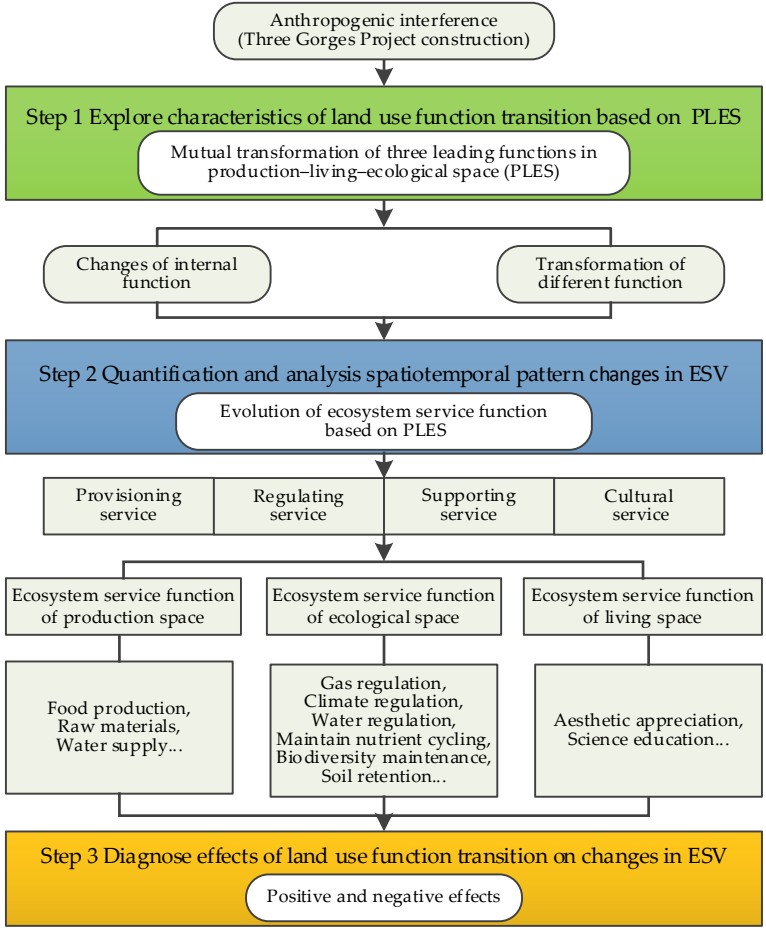

**Figure 2.** The conceptual framework of land use function transition (LUFT) and its effects on ecosystem service value (ESV).

Step 1: Explore the spatiotemporal evolution characteristics of LUFT during the different construction stages of the TGP from the perspective of PLES.

Step 2: Quantify ESV in the TGRA-HS from 1990 to 2020 using the revised assessment model and analysis of the evolution of its spatiotemporal pattern.

Step 3: Diagnose the effects of LUFT in terms of changes in ESV during the construction of the TGP in the TGRA-HS from 1990 to 2020.

### 3.2. Analysis of LUFT

#### 3.2.1. PLES Classification System

The analysis of the PLES theory and the diagnosis of the relationships between land use types and functions are essential. This study referred to the PLES classification systems used in past studies [48,49]. On the basis of the leading function, and combined with the multi-functional attributes of land use [50,51], we divided land use types into different functional types of PLES [48,52]. Additionally, it was divided into 3 primary types (including P-space, L-space, and E-space) and 8 secondary types (including APS, IPS, ULS, RLS, FES, GES, WES, and OES) (Table 1). More specifically, P-space can be divided into APS and IPS, which primarily includes cultivated land and other construction land, such as mining land and transportation land. L-space can be divided into ULS and RLS and mainly includes urban land and rural residential land. E-space can be divided into FES, GES, WES, and OES and primarily includes all land types except cultivated land, rural and urban construction land, and industrial and mining land. According to the above criteria for PLES and the land use classification system in Table 1, this study used ArcGIS10.8 to obtain the PLES of the TGRA-HS in 1990, 2000, 2010, and 2020. The evaluation and structural transform characteristics of PLES in the TGRA-HS from 1990 to 2020 were obtained so as to lay the foundation for diagnosing the effects of LUFT on changes in ESV.

**Table 1.** Corresponding table of PLES and land use type.

| Primary Type | Secondary Type | Land Use Interpretation and Classification |
|---|---|---|
| Production space (P-space ) | Agricultural production space (APS) | Paddy field, dry land |
|  | Industrial production space (IPS) | Industrial and mining land, transportation construction land |
| Living space (L-space) | Urban Living space (ULS) | Urban land |
|  | Rural Living space (RLS) | Rural residential land |
| Ecological space (E-space) | Forestland Ecological space (FES) | Forestland, shrub land, sparse forestland, other forestland |
|  | Grassland Ecological space (GES) | High-coverage grassland, medium-coverage grassland, low-coverage grassland |
|  | Water Ecological space (WES) | Canals, lakes, permanent glaciers, snowfields, tidal flats, beaches, reservoirs and pond |
|  | Other Ecological space (OES) | Sandy land, Gobi, saline–alkali land, marshland, bare land, bare rock texture, other land |

#### 3.2.2. Methods of LUFT

The transition of land use function is realized by a transition matrix, and it can calculate the amount and direction of transformation among land use function types [53]. This method is derived from the quantitative description of the system state and state transition in the system analysis, which can better describe changes in land use function type. Based on the PLES classification system in Table 1, this study adopts a transition matrix to represent the LUFT [54]. The characteristics of the PLES evaluation and structural transition in the TGRA-HS from 1990 to 2020 were obtained. The mathematical model is:

$$C_{ij} = 100A_{ij}^t + A_{ij}^{t+1} \tag{1}$$

$$C_{ij} = \begin{pmatrix} C_{11} & C_{12} & \cdots & C_{1n} \\ C_{21} & C_{22} & \cdots & C_{2n} \\ \cdots & \cdots & \cdots & \cdots \\ C_{n1} & C_{n2} & \cdots & C_{nn} \end{pmatrix} \tag{2}$$

where *i* and *j* are the two types of land use function, respectively; $C_{ij}$ is the transition matrix; and $A_{ij}^{t}$ and $A_{ij}^{t+1}$ are the coding of land use function status at time *t* and *t+1*, respectively. For example, the coding of land use function status at time *t* and *t+1* is 11 (representing APS) and 31 (representing FES), respectively, so $C_{ij}$ is 1131, which represents the conversion from APS to FES.

### 3.2.3. Time Node Division

According to the stage deployment, the construction of the TGP can be divided into stages, such as the construction preparation and start of the first phase of the project (1993), the realization of the river closure and the start of the second phase of the project (1997), the start of water storage and the completion of the second phase of the project (2003), water storage of 175m and the completion of the third phase of the project (2009), and the realization of the full-scale operation of the TGP (2015) (Figure 3). There were obvious differences in the spatial pattern of PLES and its variation trend before and after the dam construction. However, it was difficult to carry out eco-environmental monitoring and change analysis in the whole process of project construction because of the difficulty of obtaining long-term data. Therefore, this study selected 1990, 2000, 2010, and 2020 as the time nodes to explore the characteristics of LUFT, the spatiotemporal change of ESV, and the effects of LUFT on the change in ESV from the project demonstration to the full operation stage according to the construction stage division of the TGP and the availability of data.

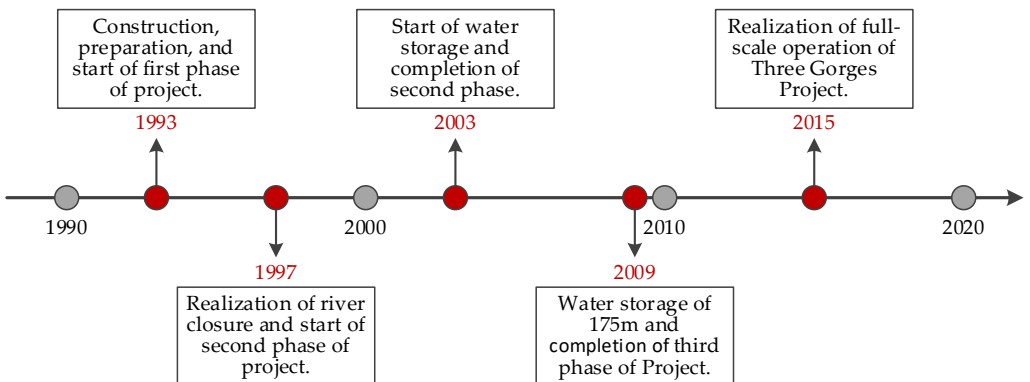

**Figure 3.** Division of the construction stage of the Three Gorges Project (TGP).

### 3.3. Assessment of ESV from the Perspective of PLES

The PLES can better reflect the comprehensive characteristics of ecosystem services. ESV is a quantitative assessment of ecosystem service capacity, and the quantitative assessment of ESV can help to transform ecological issues into indicators that are easy to understand for the public and that can help to identify problems. In this study, the equivalent factor of ESV per unit area was used to evaluate the ESV from the perspective of PLES in the TGRA-HS between 1990 and 2020. The outstanding advantage of the methods is their simplicity, and the value of ecosystem services in the form of currency. The equivalent value per unit area of APS, FES, GES, WES, and OES corresponded to the cultivated land, forestland, grassland, waters, and bare land in the existing studies, respectively. Additionally, they were modified and developed for the dynamic evaluation on Chinese terrestrial ecosystem service value. Among them, the equivalent value of APS was calculated based on the proportion composition of paddy field and dry land in the TGRA-HS. Although IPS, ULS, and RLS do not belong to natural ecological spaces, they still have various degree of ecosystem service function due to the existence of green space. This study refers to the equivalent value of construction land from the existing research [55,56] and adjusts the equivalent of different spaces according to the degree of human interference. In this way, the equivalent factor table of ecological services can be obtained (Table 2).

**Table 2.** Ecosystem service equivalent value per unit area in the TGRA-HS (unit: CNY/hm$^2$).

| Category | Provision Service | Regulating Service | Supporting Service | Cultural Service |
|---|---|---|---|---|
| APS | 145.73 | 8743.63 | 2477.35 | 233.17 |
| IPS | −29,145.40 | −32,059.95 | 408.04 | 29.15 |
| ULS | −21,859.05 | −14,222.96 | 1253.26 | 32.15 |
| RLS | −8306.44 | 1165.81 | 1748.74 | 174.87 |
| FES | 3293.43 | 37,976.45 | 13,465.18 | 2710.53 |
| GES | 2215.05 | 22,704.27 | 8510.46 | 1719.58 |
| WES | 27,163.51 | 150,910.91 | 10,346.62 | 5508.48 |
| OES | 0.00 | 437.19 | 116.58 | 35.45 |

However, the vegetation coverage affected a variety of ecological processes and played an important role in ecosystem services. Owing to the differences in vegetation coverage flourishing in the types of cultivated land, forestland, and grassland covered with vegetation, the same types of land might provide significantly different levels of ecosystem service value [57]. Therefore, this study used NDVI to reflect the vegetation flourishing status in the TGRA-HS. The ESV was revised once again. Finally, the quantity and spatial characteristics of ESV from the perspective of the PLES of each period in the construction of the TGRA-HS was calculated using the ESV model. The mathematical ESV model is:

$$ESV = \sum_{i=1}^{m} \sum_{j=1}^{n} \sum_{k=1}^{o} A_{ij} B_{ik} C_{ik} \tag{3}$$

$$C_{ik} = \frac{NDVI_{ik}}{\overline{NDVI_i}} \tag{4}$$

where $A_{ij}$ is the ESV coefficient of the $j_{th}$ ecosystem services of the $i_{th}$ PLES type; $B_{ik}$ is the area of the $i_{th}$ PLES type in the $k_{th}$ research unit; $C_{ik}$ is the revision coefficient of the $NDVI$ of the $i_{th}$ PLES type in the $k_{th}$ research unit; $\overline{NDVI_i}$ is the mean of the $i_{th}$ PLES type; $NDVI_{ik}$ is the $NDVI$ value of the $i_{th}$ PLES type in the $k_{th}$ research unit; and $i$, $j$, and $k$ are the PLES type, ecosystem services type, and number of research units, respectively. Then, a 1 km × 1 km vector grid was designed to cover the whole TGRA-HS, and it was divided into 11,612 grid units to explore spatial differences in ESV. Additionally, the ESV was divided into five levels (including slight, light, moderate, severe, and extreme levels) based on the natural breaks (Jenks) with the help of ArcGIS10.8.

*3.4. The Effects of LUFT on Changes of ESV*

The ecological contribution rate of LUFT refers to the change in the regional ESV caused by a certain land use function change. It can better identify the functional types that affect the change in ESV, which is conducive to the discussion of the leading factors of regional ecological environmental change [58]. Therefore, this study adopts the ecological contribution rate of LUFT to quantify the effects of LUFT on changes in regional ESV in the TGRA-HS. The mathematical model is [37]:

$$ESV_C = \frac{(ESV_{t+1} - ESV_t) \times CA}{TA} \tag{5}$$

where $ESV_t$ and $ESV_{t+1}$ are the ESV reflected by a certain LUFT at time $t$ and $t+1$, respectively; $ESV_C$ is the contribution rate of LUFT; $CA$ is the transition area; and $TA$ is the total area of the study area.

*3.5. Data Sources*

Data were collected from multiple sources (Table 3). The data on PLES in the TGRA-HS in 1990, 2000, 2010, and 2020 were taken from land use data, including 6 primary types and 25 secondary types, which were downloaded from the Geospatial Data Cloud. They were extracted at a resolution of 30 m × 30 m to explore the characteristics of

LUFT and the evolution of the ESV change, according to the construction stage division of the TGP. Meanwhile, the administrative map of the TGRA-HS was taken from the department of natural resources of the Hubei Province. Furthermore, the economic and social development data were taken from the Hubei Statistical Yearbook from 1990 to 2020 [59]. The geodatabase of the TGRA-HS was built in the ArcGIS10.8 software platform to process the above data.

**Table 3.** Data and sources.

| Data | Sources |
|---|---|
| Land use data | Geospatial Data Cloud |
| Digital elevation model data | (http://www.gscloud.cn, accessed on 17 January 2023) |
| Administrative map | Department of Natural Resources of Hubei Province (https://zrzyt.hubei.gov.cn/, accessed on 17 January 2023) |
| Normalized difference vegetation index (NDVI) data | Resource and Environment Science and Data Center of the Chinese Academy of Sciences (http://www.resdc.cn/, accessed on 17 January 2023) |
| Economic and social development data | Hubei Statistical Yearbook |

## 4. Results

### 4.1. Characteristics of the Transition of Land Use Function

4.1.1. Evaluation of Land Use Function in the TGRA-HS from 1990 to 2020

The spatiotemporal patterns of PLES were basically the same in the TGRA-HS, and they remained consistent as a whole in 1990, 2000, 2010, and 2020 (Table 4). The evolution of PLES showed characteristics of a reduction in P-space, and the expansion of L-space and E-space from 1990 to 2020. The structure of PLES was mainly dominated by FES, which accounted for over 77% of the total area of the TGRA-HS with an absolute dominant position during the study period (Figure 4). The P-space was mainly dominated by APS, presenting the characteristic of aggregated distribution, and its proportion was reduced from 12.86% to 11.59%. It was concentrated in the form of mass shape and mainly distributed in low-lying areas below 500 meters in the southeast and scattered in the west of the TGRA-HS in the form of point shapes. The E-space was mainly dominated by FES, which also presented the characteristic of aggregated distribution, and its proportion was reduced from 78.80% to 77.71%. It was mainly distributed in the mountain zone of the TGRA-HS. Meanwhile, the L-space was mainly dominated by ULS, presenting the characteristic of dotted distribution, and its proportion increased from 0.31% to 0.57%.

**Table 4.** The area proportion and change of production−living−ecological space (PLES) in the TGRA-HS between 1990 and 2020 (unit: %).

| Years/Period | P-space | | L-space | | E-space | | | |
|---|---|---|---|---|---|---|---|---|
| | APS | IPS | ULS | RLS | FES | GES | WES | OES |
| 1990 | 12.86 | 0.04 | 0.31 | 0.19 | 78.80 | 6.70 | 1.10 | 0.00 |
| 2000 | 12.85 | 0.32 | 0.39 | 0.20 | 78.44 | 6.70 | 1.11 | 0.00 |
| 2010 | 11.81 | 0.52 | 0.56 | 0.24 | 78.11 | 6.82 | 1.94 | 0.00 |
| 2020 | 11.59 | 1.12 | 0.57 | 0.24 | 77.71 | 6.80 | 1.96 | 0.00 |
| 1990–2000 | −0.02 | 0.28 | 0.08 | 0.01 | −0.35 | 0.00 | 0.00 | 0.00 |
| 2000–2010 | −1.04 | 0.21 | 0.17 | 0.04 | −0.33 | 0.12 | 0.83 | 0.00 |
| 2010–2020 | −0.22 | 0.60 | 0.01 | 0.01 | −0.40 | −0.01 | 0.02 | 0.00 |
| 1990–2020 | −1.28 | 1.08 | 0.26 | 0.06 | −1.09 | 0.11 | 0.86 | 0.00 |

The expansion areas of ULS were concentrated in the southeast of the TGRA-HS and scattered throughout the rest of downtown Yichang. The spatial distribution of RLS was basically the same with the characteristics of agricultural production activities. It indicated that the regions that were suitable for agricultural production with relatively good natural conditions were usually suitable for harmonious human settlement. The regions had been developed into rural residential agglomerations gradually, and they remained relatively

stable. The evolution process of PLES showed the characteristics of two types down and five types up in the TGRA-HS from 1990 to 2020. The largest increase was in IPS, and the largest reduction was in APS, with a proportion of 1.08% and −1.28%, respectively. The evolution of PLES showed the characteristics of a reduction in P-space and the expansion of L-space and E-space from 1990 to 2010. This was mainly reflected in the continuous decline in APS and the continuous increase in ULS, RLS, and WES. While the P-space and L-space showed a trend of expansion, and the E-space mainly showed a trend of reduction from 2010 to 2020. The characteristics of the PLES evolution were closely related to the rapid development. The expansion of construction land had encroached on the land with relatively low economic benefits, which led to an increase in IPS, ULS, and RLS and a reduction in APS and FES.

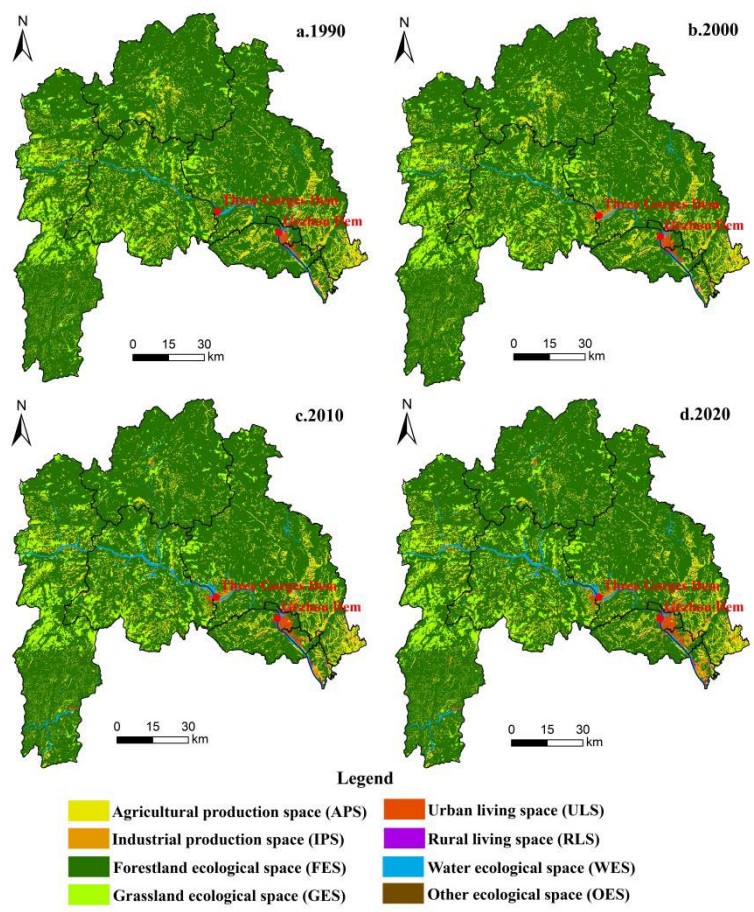

**Figure 4.** Characteristics of PLES in the TGRA-HS between 1990 and 2020.

4.1.2. Structural Transform of Land Use Function in the TGRA-HS from 1990 to 2020

For a further intuitive analysis, the structural transformation of the land use function in the TGRA-HS from 1990 to 2010 was calculated based on the mathematical model of LUFT. Compared with the data of the base period in 1990, the IPS and ULS increased significantly, and the growth rate was more than 85% during the study period (Table 5). Meanwhile, the FES and GES only showed minimal changes, and they were almost the same as the data of the base period. The differences in the spatial distribution characteristics of structural transform in different phases were obvious and were mainly reflected in the conversion of P-space and E-space in the TGRA-HS. From 1990 to 2000, the mutual conversion area was 6679.08 hm$^2$, which was mainly reflected in the conversion of APS, FES, and IPS, with a proportion of 70.84% of the total converted area. At this stage, the TGP and the resettlement of immigrants in the reservoir area had been implemented on a grand scale, resulting in the conversion of a large number of FES into IPS and APS. From 2000 to 2010, the mutual conversion area was 37,480.27 hm$^2$, which was mainly reflected in the conversion of APS

and FES, accounting for 66.88% of the total converted area during this period. The Grain for Green Project (GGP) was implemented, which resulted in a large amount of cultivated land being converted to forestland. At this stage, the Three Gorges Dam was officially closed for water storage, and the water level rose from 135 m to 175 m. This led to the transformation of an important proportion of APS into WES. Meanwhile, the mutual conversion area was 5916.48 hm$^2$ from 2010 to 2020, which was mainly reflected in the conversion of FES and IPS, accounting for 78.06% of the total converted area in the TGRA-HS. They were mainly concentrated in the urban area of Yichang around the Three Gorges Dam and the Gezhou Dam (Figure 5).

**Table 5.** Ordered Tupu table of land use transition in the TGRA-HS between 1990 and 2020 calculated based on Equations (1) and (2).

| Period | Number | Transition Types | Area (hm$^2$) | Change Rates (%) | Accumulative Change Rates (%) |
|---|---|---|---|---|---|
| 1990–2000 | 1 | FES → IPS | 2507.81 | 37.55 | 37.55 |
| | 2 | FES → APS | 1602.92 | 24.00 | 61.55 |
| | 3 | APS → FES | 620.96 | 9.30 | 70.85 |
| | 4 | APS → ULS | 548.97 | 8.22 | 79.07 |
| | 5 | FES → ULS | 352.47 | 5.28 | 84.34 |
| | 6 | WES → IPS | 287.52 | 4.30 | 88.65 |
| | 7 | WES → APS | 172.02 | 2.58 | 91.22 |
| | 8 | APS → GES | 134.79 | 2.02 | 93.24 |
| | 9 | APS → WES | 124.72 | 1.87 | 95.11 |
| | 10 | FES → RLS | 86.41 | 1.29 | 96.40 |
| 2000–2010 | 1 | APS → FES | 16,149.51 | 43.09 | 43.09 |
| | 2 | FES → APS | 8917.54 | 23.79 | 66.88 |
| | 3 | FES → IPS | 2452.59 | 6.54 | 73.43 |
| | 4 | APS → GES | 2415.21 | 6.44 | 79.87 |
| | 5 | APS → WES | 1544.04 | 4.12 | 83.99 |
| | 6 | FES → ULS | 907.02 | 2.42 | 86.41 |
| | 7 | IPS → ULS | 778.33 | 2.08 | 88.49 |
| | 8 | APS → ULS | 699.77 | 1.87 | 90.35 |
| | 9 | IPS → WES | 532.49 | 1.42 | 91.77 |
| | 10 | FES → RLS | 498.83 | 1.33 | 93.11 |
| 2010–2020 | 1 | FES → IPS | 4618.55 | 78.06 | 78.06 |
| | 2 | APS → FES | 219.98 | 3.72 | 81.78 |
| | 3 | FES → APS | 182.07 | 3.08 | 84.86 |
| | 4 | FES → ULS | 136.73 | 2.31 | 87.17 |
| | 5 | WES → IPS | 132.18 | 2.23 | 89.40 |
| | 6 | APS → RLS | 115.94 | 1.96 | 91.36 |
| | 7 | IPS → FES | 85.78 | 1.45 | 92.81 |
| | 8 | APS → ULS | 71.00 | 1.20 | 94.01 |
| | 9 | FES → RLS | 58.15 | 0.98 | 94.99 |
| | 10 | RLS → APS | 54.83 | 0.93 | 95.92 |

From the perspective of the structure characteristic of the PLES transform-in from 1990 to 2020, the APS was mainly converted from FES, and the proportion was 90.86%. The IPS was mainly converted from FES, and the proportion was as high as 93.11%. The ULS and RLS were mainly converted from APS and FES, while FES, GES, and WES were mainly converted from APS. From the perspective of the structural characteristics of the PLES transform-out from 1990 to 2020, the APS was mainly converted to FES, and the proportion of transition was 72.82%. The IPS was mainly converted to ULS, with a proportion of more than 82.49% of the total transition of this type. The RLS was mainly converted to IPS, with a transition ratio of 43.76%. The FES was mainly converted to APS and IPS. The GES was mainly converted to APS, and the WES was mainly converted to APS and IPS. The structural transform characteristics of PLES in different phases reflected the transformation of socio-economic development in the TGRA. We should pay more attention to the improvement of

the quality and the optimization of the structure. The win–win situation of PLES will be realized through the optimization of the structure in the whole area.

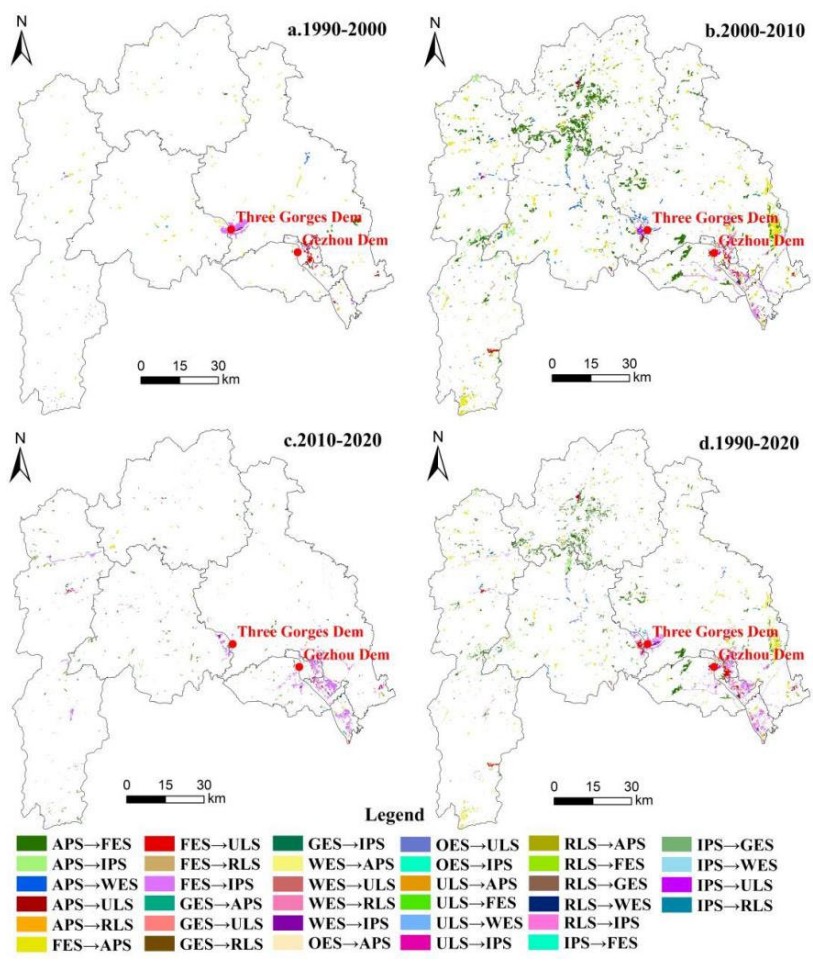

**Figure 5.** Spatial pattern evolution of PLES in the TGRA-HS between 1990 and 2020.

*4.2. Characteristics of ESV Change*

4.2.1. Quantity Characteristics of ESV Change from the perspective of PLES

According to Equations (3) and (4), the ESV of each spatial type in the TGRA-HS was calculated from 1990 to 2020 (Table 6). We could see that the total ESV of the PLES of the TGRA-HS in 1990, 2000, 2010, and 2020 was CNY $619.22 \times 10^8$, CNY $614.43 \times 10^8$, CNY $628.51 \times 10^8$, and CNY $621.39 \times 10^8$, respectively. The total ESV of the PLES showed a trend of floating change in the TGRA-HS, but remained relatively stable as a whole. Among them, the E-space contributed the most to the total ESV, with a proportion of more than 90% in different phases. The total ESV of E-space rose to a certain extent, increased by CNY $13.06 \times 10^8$. The total ESV of P-space and L-space decreased significantly from 1990 to 2020, with a value of CNY $-9.73 \times 10^8$ and CNY $-1.15 \times 10^8$, respectively. From the perspective of the corresponding secondary types, the total ESV of IPS and FES declined the most during the past 30 years, with a value of CNY $7.93 \times 10^8$ and CNY $7.56 \times 10^8$, respectively. The total ESV of APS, ULS, and RLS also declined significantly, and the total ESV of WES and GES increased by CNY $20.17 \times 10^8$ and CNY $0.45 \times 10^8$, respectively. At the same time, the GGP since 1998 has accelerated the improvement of ESV.

**Table 6.** Changes of ESV of PLES in the TGRA-HS between 1990 and 2020 calculated based on Equation (3) and (4) (unit: CNY $10^6$).

| Years/Period | P-space | | L-space | | E-space | | | |
|---|---|---|---|---|---|---|---|---|
| | APS | IPS | ULS | RLS | FES | GES | WES | OES |
| 1990 | 1807.98 | −29.96 | −130.27 | −11.71 | 54842.66 | 2852.42 | 2590.48 | 0.03 |
| 2000 | 1805.77 | −232.62 | −164.14 | −12.54 | 54597.45 | 2852.05 | 2596.65 | 0.02 |
| 2010 | 1660.12 | −384.79 | −236.39 | −14.91 | 54366.90 | 2902.79 | 4556.84 | 0.00 |
| 2020 | 1628.64 | −823.90 | −241.98 | −15.41 | 54086.03 | 2897.82 | 4607.67 | 0.00 |
| 1990−2000 | −2.21 | −202.66 | −33.87 | −0.83 | −245.21 | −0.37 | 6.17 | −0.01 |
| 2000−2010 | −145.64 | −152.17 | −72.25 | −2.37 | −230.55 | 50.75 | 1960.20 | −0.02 |
| 2010−2020 | −31.49 | −439.11 | −5.59 | −0.50 | −280.87 | −4.98 | 50.83 | 0.00 |
| 1990−2020 | −179.34 | −793.94 | −111.71 | −3.70 | −756.62 | 45.40 | 2017.20 | −0.03 |

4.2.2. Spatial Characteristics of ESV Change

According to the calculation results, the spatial characteristics of the ESV change in the TGRA-HS between 1990 and 2020 were obtained (Figure 6). The spatial pattern of the slight ESV area in the TGRA-HS was mainly distributed in the urban built-up area. It showed an obvious expansion trend with the development of urbanization and the construction of the TGP. The distribution pattern of the light and moderate ESV area was consistent with the spatial distribution of cultivated land, and most of them had been replaced by slight areas during the past 30 years. The spatial pattern of the severe ESV area was basically the same as the distribution of forestland. Additionally, the characteristics of spatial distribution tended to be fragmented from 1990 to 2020, which was related to the encroachment of forestland by construction land. The distribution pattern of extreme ESV area was consistent with the river. The population distribution in this region was relatively concentrated along the river banks with deep valleys and steep slopes, and the area of sloping farmland was large. People's daily life and agricultural production activities had destroyed the vegetation along the coast, causing serious water and soil loss.

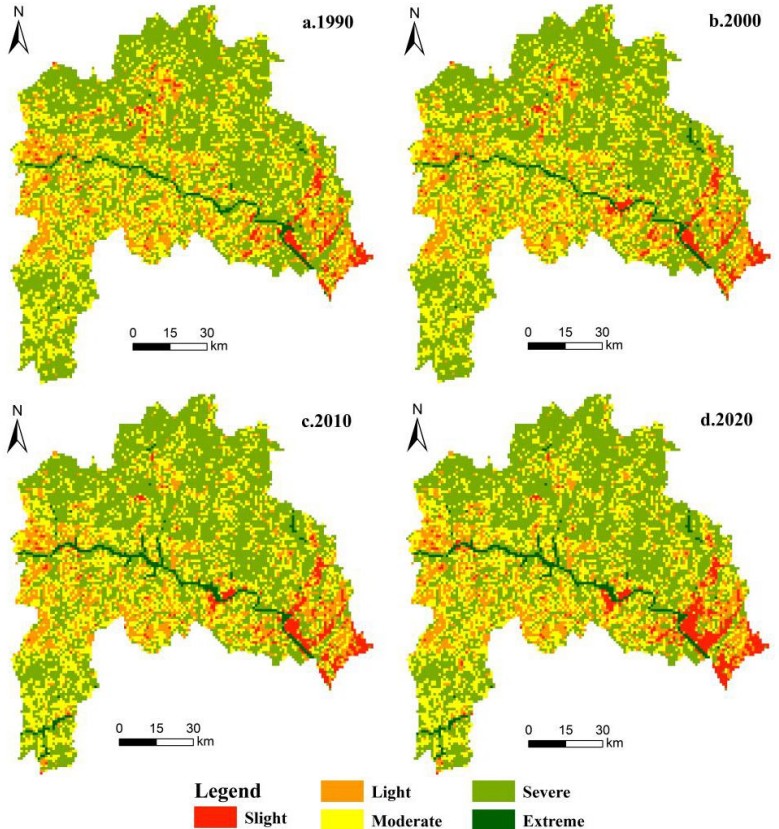

**Figure 6.** Spatial evolution of ESV in the TGRA-HS between 1990 and 2020.

### 4.3. The Effects of LUFT on Changes of ESV

According to Equation (5), the effects of LUFT on changes in ESV were calculated (Table 7). We could see that the transformation of PLES had brought about a decline in ESV in the TGRA-HS from 1990 to 2020. This mainly stemmed from the encroachment of P-space on E-space. The transition of E-space to P-space had the greatest impact on the reduction in ESV, and the proportion of contribution rates accounted for 82.76%. The transition of E-space to L-space was far less than that of P-space, and the proportion of the contribution rate was 12.68%. From the corresponding secondary type, the main reason for the decline in ESV in the TGRA-HS, stemmed from the encroachment of IPS and APS on FES, and the proportion of contribution rates in negative effects was 52.37% and 23.47%, respectively. Meanwhile, the transformation of PLES brought about an increase in ESV in the TGRA-HS from 1990 to 2020. This mainly stemmed from the encroachment of E-space on P-space. The transition of P-space to E-space had the greatest impact on the increase in ESV in the TGRA-HS, and the proportion of contribution rates accounted for 94.45%. From the corresponding secondary type, the main reason for the increase in ESV in the TGRA-HS stemmed from the increase in FES, WES, and GES, and the proportion of contribution rates in positive effects was 64.33%, 24.98%, and 5.14%, respectively.

**Table 7.** Main types and contribution rates of LUFT in the TGRA-HS between 1990 and 2020 calculated based on Equation (5).

| Effect Type | Transition Type | Difference of ESV (CNY $10^6$) | Contribution Rates (%) | Proportion of Contribution Rates (%) |
|---|---|---|---|---|
| Positive effects of ESV | APS → FES | 747.41 | 0.358317 | 64.33 |
| | APS → WES | 290.21 | 0.139130 | 24.98 |
| | APS → GES | 59.70 | 0.028619 | 5.14 |
| | ULS → WES | 29.70 | 0.014240 | 2.56 |
| | RLS → WES | 14.25 | 0.006832 | 1.23 |
| | ULS → FES | 4.35 | 0.002085 | 0.37 |
| | IPS → ULS | 3.91 | 0.001875 | 0.34 |
| | RLS → APS | 3.78 | 0.001812 | 0.33 |
| | RLS → FES | 3.62 | 0.001738 | 0.31 |
| | IPS → FES | 3.03 | 0.001454 | 0.26 |
| Negative effects of ESV | FES → IPS | −1016.67 | −0.487402 | 52.37 |
| | FES → APS | −455.63 | −0.218434 | 23.47 |
| | FES → ULS | −160.92 | −0.077147 | 8.29 |
| | WES → IPS | −69.08 | −0.033119 | 3.56 |
| | APS → ULS | −61.52 | −0.029492 | 3.17 |
| | WES → APS | −46.73 | −0.022405 | 2.41 |
| | WES → ULS | −40.87 | −0.019594 | 2.11 |
| | FES → RLS | −39.45 | −0.018914 | 2.03 |
| | RLS → IPS | −15.30 | −0.007337 | 0.79 |
| | GES → APS | −11.94 | −0.005724 | 0.61 |
| | APS → RLS | −10.64 | −0.005100 | 0.55 |

In summary, the LUFT caused by human construction projects affects the spatiotemporal changes in the regional ESV, and the trends of the improvement and deterioration of ecosystem services can offset each other on the regional scale to keep the ESV relatively stable as a whole. However, the stability of the ESV does not mean that the eco-environment has not changed. The enhancement of LPS, IPS, and APS in the study area was the main reason for the decrease in ESV. Meanwhile, the improvement of ecosystem service functions was mainly achieved via the conservation of E-space and the GGP since 1998 in the TGRA. This shows that maintaining the stability of forestland, water land, and grass land ecosystems was crucial in improving the ESV in the TGRA. In addition, the transformation of L-space into E-space and P-space in the TGRA-HS also led to the improvement of ecosystem services. This is mainly due to the consolidation of rural settlements in recent years, but the proportion of contribution rates was relatively low.

## 5. Discussion

Existing research is more concerned with the calculation of ESV from the perspective of land use types, which may not reveal the mutual feedback relationship between anthropogenic interference and ecosystem services well. The PLES is a comprehensive method of dividing territorial space that can better reflect the comprehensive characteristics of ecosystem services with different land use types, and its layout and evolution have a profound impact on the regional ecosystem function. Taking the TGRA-HS as a study area, we explored the characteristics of LUFT, the spatiotemporal change of ESV, and the effects of LUFT on the change in ESV from the perspective of PLES by following the construction stage division of the TGP. This study revealed the mutual feedback relationship between human social systems and ecosystems well, making up for the shortcomings of the existing research.

### 5.1. Construction of the TGP and Stage Response

According to the stage deployment, the construction of the TGP can be divided into different stages, and there were obvious differences in the spatial pattern of PLES and its variation trend. The characteristic of LUFT was consistent with the period of project construction, national policies, and regional development plans in the TGRA-HS. Human activities, such as facility construction and resettlement, had a great impact on the structure and function of land use in the reservoir area. Before and after the construction of the TGP, the changes in the water level of the main stream of the Yangtze River were the most intuitive manifestation of the project's impact on the LUFT.

During the construction preparation and the start of the first phase of the project, the disturbance of the project was still small, and the LUFT was mainly driven by regional agricultural development planning and the early resettlement policy. In 1993, the resettlement of the TGRA began to be carried out on a large scale. The early resettlement strategy was local resettlement and local construction. To obtain food and income, the agricultural population reclaimed the sloping land spontaneously. In 2003, the TGP officially closed the sluice and stored water. The large-scale land use transition during this period was mainly driven by project water storage. The transformation of land use function mainly occurs in river valleys below 500 meters. The main reason for the increase in WES was that the reservoir area begun to store water after the Three Gorges Dam was built. In addition, driven by national policies such as the adjustment of resettlement policy and the pilot program of the "Two-oriented Society", FES and GES were restored and IPS and ULS were expanded. After 2010, the reservoir entered the stage of full operation, and the changes in land use patterns brought about by the construction of the project tended to be stable. Driven by national policies and regional development planning, the major functional areas divided the three western counties into key ecological function zones at the national level and restricted their development. The five eastern districts were key development zones at the provincial level with rapid urban development [60]. The development drew lessons from the past, and the economic development and ecological protection were more focused on diversity and coordination so as to achieve sustainable development. The project construction also brought about some policy responses to a certain extent. During the construction of the TGP, the country had successively implemented key ecological projects in the TGRA to mitigate the negative effects of the project. The territorial space ecological security system with forest vegetation dominated, and a combination of forest and grass was initially established to ensure the eco-environmental safety of the reservoir area and the safety of the reservoir operation. This also shows that maintaining the stability of forestland, water land, and grass land ecosystems was crucial to improving the ecosystem service function in the TGRA-HS.

### 5.2. Implications for Theory and Practice

This study put forward the LUFT from the perspective of PLES on the basis of the project construction stage division and the revised assessment methods of ESV. The quan-

titative assessment of ESV transforms ecological issues into indicators that are easily understood by the public, and which can help identify problems [61]. With the continuous deepening of ESV and function research, the quantitative assessment of ESV is becoming more and more mature [62,63]. In this study, we refer to the existing research of Xie et al. [64] to evaluate ESV in the TGRA-HS. It is generally assumed that the same type of land use has the same value of ecosystem services, but this ignores the potential impact of vegetation flourishing. However, the vegetation affects a variety of ecological processes, and ESV assessment that ignores vegetation growth will be very inaccurate. Therefore, this study relies on remote sensing images to determine the actual status of regional vegetation and uses the NDVI to revise the ESV realistically so as to obtain the accounting result of ESV with higher accuracy. At present, the revision methods of ESV are not unified, which deserves further discussion.

Meanwhile, this study provides a scientific reference to support and serve the layout optimization of the spatial development pattern and eco-environmental protection in the TGRA-HS. Additionally, it provides an important reference for understanding the effects of major conservancy projects on the ecosystem in reservoir areas, such as Bratsk Reservoir, Samara Reservoir, Smallwood Reservoir, Lake Guri, etc. The results are helpful in creating an incentive for people to understand ecosystem services and for policymakers to use, and they can effectively assist ecological restoration across the planning area [65]. Considering that project construction is an important but irreversible process affecting ESV in the reservoir area, policymakers should focus on the following recommendation during the construction process of large-scale conservancy projects. On the one hand, they should design reasonable targets for ecological protection and construction, design feasible plans and specific measures to promote the restoration of ecosystems and enhance the ecosystem service function so as to lay the ecological foundation for regional ecological security. On the other hand, they should strengthen the construction of projects related to ecological environmental construction and protection in the reservoir area, promote the smooth implementation of vegetation restoration and ecological corridor construction projects in the reservoir area, and improve the construction standards and ecological compensation.

*5.3. Limitations and Future Works*

Exploring the influencing mechanism of regional ecosystem services is helpful in guiding regional ecological construction. The research on the influencing factors of ESV can be roughly divided into two categories. One analyzes the spatiotemporal variation characteristics of ESV based on the transformation of land use patterns. The other is to select relevant factors, establish an index system, and use regression analysis or correlation analysis to discuss the main influencing factors of ESV. Additionally, the influencing factors can be divided into natural factors (including biological, climate, soil, and topography) and human factors (including land use transition and socio-economic development). In terms of human factors, changes in land use type, overall land use pattern and land development intensity will affect the level of ecosystem services [66]. Among them, the transition of land use function induced by anthropogenic interference (such as large-scale construction projects) is one of the most important influencing factors of ecosystem services, which will directly affect the change in ESV and the flow and interaction between ecosystem services. Due to the limitations of some basic data and research objects, this study mainly explored the effects of LUFT on changes in ESV during the construction of the TGP at the macro level. However, the driving mechanism of other potential factors on ESV evolution and the stability of the ecosystem have not been explored in depth, which is also an interesting direction for ecosystem service research. It is necessary to further study the driving mechanism of ESV evolution at each stage combined with the stages of economic and social development and to reveal the related issues of land use transition. In order to more accurately grasp the response of regional ecological environments to socio-economic development, the analysis of the driving mechanism needs to be further refined and quantified in the future.

## 6. Conclusions

(1) The transition of land use function from the perspective of PLES is the mapping of the evolution of the human–nature relationship in the spatial pattern, which reflects the evolution of the spatial pattern caused by human interference with the continuous development of society.

(2) From 1990 to 2020, the distribution pattern of PLES in the TGRA-HS was basically the same, and it was mainly dominated by FES; meanwhile, there were significant regional differences. The evolution of PLES showed the characteristics of a reduction in P-space and an expansion in L-space and E-space. The mutual transformation between P-space and E-space was the main form of PLES transformation in the TGRA-HS. The characteristics of structural transformation in different phases were obvious, which reflected the phases in the construction of the TGP and transformations brought about by socio-economic development.

(3) From 1990 to 2020, the total ESV showed a trend of floating change in the TGRA-HS but remained relatively stable as a whole. The E-space contributed the most to the total ESV, and increased by CNY $13.06 \times 10^8$. The IPS and FES declined the most during the past 30 years, with a decrease of CNY $7.93 \times 10^8$ and CNY $7.56 \times 10^8$, respectively. The LUFT caused by human project construction affects the spatiotemporal change of regional ESV. The spatial pattern of the slight ESV was mainly distributed in the urban built-up area, and it showed an obvious expansion trend with the development of urbanization and the construction of the TGP.

(4) The change in regional ESV induced by LUFT reveals the whole dynamic process of the positive and negative effects of human activities on ecosystem services, and the two effects offset each other to keep the ESV relatively stable as a whole. The main reason for the decline in the ESV stemmed from the encroachment of IPS and APS in FES, whose contribution rates were 52.37% and 23.47%, respectively. The TGRA should formulate differentiated spatial governance measures based on the in-depth implementation of the main functional zone strategy so as to promote the rational layout and integrated development of the PLES in the future.

**Author Contributions:** Conceptualization, F.P. and Q.W.; methodology, F.P., N.S. and Q.H.; investigation, all authors; resources, Q.W. and N.S.; writing—original draft preparation, F.P. and Q.W.; writing—review and editing, all authors. All authors have read and agreed to the published version of the manuscript.

**Funding:** This research was funded by the National Natural Science Foundation of China, grant number 42001229; the Research Project of Philosophy and Social Sciences in Colleges and Universities of Hubei Province, grant number 21Q093; Humanity and Social Science Key Program Foundation of Ministry of Education in China, grant number 18YJC790153; and the Science Research Foundation of Wuhan Institute of Technology, grant number K2021051.

**Conflicts of Interest:** The authors declare no conflict of interest.

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
