# Peer review of "Land Use Function Transition and Associated Ecosystem Service Value Effects Based on Production–Living–Ecological Space: A Case Study in the Three Gorges Reservoir Area"

_land, doi:10.3390/land12020391_

Round 1
Reviewer 1 Report
hrough a study case in Three Gorges Reservoir area of Hubei section, China, the goal is to get to a collection of evidence by applying the effects on ecosystem service value framework they have created in the perspective of production-living-ecological classification system and, in the end, give the readers a set of policy recommendations and a best practice for real. The work is written very well. Aside from a few typos and a few sentences that look hard to read, by and large, the research design is appropriate. The introduction does provide a complete background, and the methodology is well described.
Minor flaws are as follows:
1、There are a few typos, such as Figure 2 step 1 (Explore characteristics of land use function transition based on PLES). Check out for more.
2、The article offers some good points for reflection, and the figures look of genuine interest to readers. Nevertheless, some details should be expanded, for instance, Figure 1. The marks in the figure should be complete, such as the area represented by the purple area?
3、The classification system of Table 1. Land use can be divided into production, living and ecological function on the basis of the leading functions and using types. The first class is similar to the classification standard in 2007 including 6 primary types and 25 secondary types. The classification and naming of the second class have the characteristics of the region. The naming should be analyzed or explained (or described in the paper first). Therefore, it is recommended to make necessary analysis for the classification and add references.
4、The land use function transition is the further deepening and application of the land use theory in the research of territorial space in the new era. The specific process or basis of land use function transition mathematical model need to be explained by further reference.
Overall, this paper is of great significance and addresses a relevant theme for practice. Good luck, hope to see the article published soon.
Reviewer 2 Report
About the paper with the title "Transition of Land Use Function and Ecosystem Service Value Based on Production-Living-Ecological Space: A Case Study of the Three Gorges Reservoir Area of Hubei Section, China" I have the following comments:
I suggest to find a short, but more objective title. In addition, I suggest to rewrite the abstract in a more objective and clear way. For example, I found difficulty to understand the following sentences from the abstract: "The trend of eco-environmental improvement was slightly higher than the trend of degradation in the TGRA-HS. The main reasons for the decline of ESV stemmed from the encroachments of industrial production space...". Be clear also about objectives, gaps, methodologies, novelties and main insights.
Be clear about the sources of the data and justify scientifically the methodologies considered and why they are more adjusted than other appoaches available in the literature.
The results obtained need to be better explained and clearly linked with the methodology considered. For example, be clear how did you obtain the results presented in tables 3, 4, 5 and 6. Connect these results with the models presented. I suggest you present in each tables the model used to obtain the respective results.
Reviewer 3 Report
In my opinion, the paper is interesting, it is well documented and structured. The results are supported by data and discussed accordingly.
I think that the paper fulfills the conditions of a good scientific paper and I recommend accepting the it for publication.
However, I found that not all formulas and methods used have references. They must have references if they are not original.
Round 2
Reviewer 2 Report
Accept in present form.